# Study on Symmetry and Asymmetry Rolling of AA2519-T62 Alloy at Room-Temperature and Cryogenic Conditions

**DOI:** 10.3390/ma15217712

**Published:** 2022-11-02

**Authors:** Robert Kosturek, Sebastian Mróz, Andrzej Stefanik, Piotr Szota, Piotr Gębara, Agata Merda, Marcin Wachowski, Michał Gloc

**Affiliations:** 1Faculty of Mechanical Engineering, Military University of Technology, 2 Gen. S. Kaliskiego Str., 00-908 Warsaw, Poland; 2Faculty of Production Engineering and Materials Technology, Częstochowa University of Technology, Armii Krajowej 19 Av., 42-201 Częstochowa, Poland; 3Faculty of Materials Science and Engineering, Warsaw University of Technology, 141 Woloska Str., 02-507 Warsaw, Poland

**Keywords:** AA2519, aluminum, cryogenic, asymmetry rolling, cold working, microstructure

## Abstract

The aim of this investigation was to identify the effect of rolling at room temperature and under cryogenic conditions on selected properties and the microstructure of the AA2519-T62 aluminum alloy. The rolling processes were conducted with different variants of asymmetry (1.0—symmetry rolling; 1.2, 1.4 and 1.6). The investigation of the obtained samples involves microhardness distribution, microstrains, and microstructure observations using light and transmission electron microscopes. Both rolling at room temperature and under cryogenic conditions increased the micro-hardness of AA2519-T62 by at least 10%. The highest reported increase (25%) was obtained for the sample rolled at room temperature in the symmetry rolling process. The samples rolled under cryogenic conditions are characterized by a lower increase in microhardness than samples rolled at room temperature and by significantly lower values of microstrains. The application of rolling with the asymmetry ratio remaining within the range of 1.2–16 only slightly affected the microhardness values of the samples rolled at room temperature and under cryogenic conditions with respect to conventional symmetrical rolling.

## 1. Introduction

By combining low density and high strength, aluminum alloys are invaluable construction materials in automotive and aerospace applications. In recent years a lot of scientific effort has been put into improving the mechanical properties of these materials by different techniques, such as severe plastic deformation (SPD) [1,2,3] and deformation under cryogenic conditions, in its most popular form of cryogenic rolling (CR) [4,5]. Although both of these techniques result in significant strain hardening in the case of CR, besides grain refinement, we obtain a very high density of dislocations in a deformed material, entailing greater hardening. During SPD, along with increasing strain the rate of defect storage in a material decreases, and the phenomena of dynamic recovery and recrystallization occur, preventing further structural refinement [6]. The main idea of introducing a deformation under cryogenic conditions is to inhibit all processes of dynamic recovery in worked metal, facilitating the formation of a fine-grained structure [7,8]. The grain refinement will increase the participation of grain boundaries in a material’s structure, which are an obstacle to the dislocation movement, and, as a result, it will improve an alloy’s strength [9].

An important issue in the development of new, high-strength aluminum alloys is increasing the mechanical properties of precipitation-hardened alloys. In their basic condition, these alloys contain a strengthening phase in their structure, which is responsible for their high strength properties [10,11,12]. Excellent examples of these materials are AA2219-T87 (UTS = 470 MPa) and AA7075-T651 (UTS = 570 MPa) alloys, widely used in the aerospace and military industry [13,14]. Examples of these applications include aircraft [15], sub-caliber ammunition [16], military bridges [17], and missile components [18]. Additional metal forming makes it possible to increase their strength; however, appropriate process conditions should be maintained, especially low temperature, to prevent the overaging of the strengthening phase, which would result in a drastic decrease in an alloy’s strength [19]. Thus, deformation under cryogenic conditions seems to be a good choice which allows a greatly strengthened alloy to be obtained, via both precipitates and grain boundaries. Additionally, these alloys generally have relatively narrow operating temperature ranges, avoiding the risk of recrystallization of the highly defected grainy structure [20]. Some works on the deformation under cryogenic conditions of these materials can be found in the literature [7,8,21,22,23,24,25].

One relatively new, interesting alloy, a modification of AA2219, is AA2519 [25]. It is used as a construction material in light military vehicles and is characterized by high specific strength and good ballistic properties [26,27]. In the Institute of Non-Ferrous Metals in Gliwice (Poland), a new type of this alloy has been developed, modified by an addition of scandium and zirconium, and is used as a base material in this investigation [28]. A possibility to increase the mechanical properties of high-strength aluminum alloys, including AA2519, by plastic deformation under cryogenic conditions has been described in the literature. Azimi et al. investigated the dynamic mechanical properties of AA2519 processed via multi-axial cryogenic forging and reported the increase in dynamic flow stress in cryoforged alloy [8]. Additionally, it has been observed the aluminum matrix of the cryoforged alloy is more densely populated with fragmented particles than the alloy forged at room temperature [8]. Zuiko et al. examined the strengthening mechanisms during the cryogenic rolling of solutionized AA2519 alloy and found that the strengthening effect of the cryogenic rolling is mainly contributed by the dislocation and the hardening of the low-angle boundaries [29]. Katakam et al. subjected AA2219 alloy to cryorolling and achieved a 13% and 18% increase in tensile strength and yield strength, respectively [22]. Das et al. focused on the mechanical properties of cryorolled AA7075 and reported that the rolled samples with 70% thickness reduction increased their yield and tensile strength from 540 MPa to 560 MPa, and 550 MPa to 573 MPa, respectively [23].

The asymmetry rolling process has been well known for many years, although it is still being modified [30,31]. The introduction of asymmetry in the form of a difference in the peripheral speeds or diameters of the rolls lowers the force of rolling and, as a result of the improvement of the sheet flatness, reduces the dimensional deviations in their length and width. However, in recent years, the asymmetry rolling process has also been analyzed in terms of an improvement of the properties of the rolled products [32,33]. For example, the use of the asymmetry rolling process for austenitic steels intensified the recrystallization and refinement of the structure in the central layer for temperatures above 900 °C. In turn, for rolling temperatures below 900 °C, the phenomenon of dynamic recrystallization that occurs in the asymmetrical process was not observed in a symmetrical rolling process even with the use of a 60% rolling reduction.

A significant effect of asymmetrical rolling on the mechanical properties was also observed in sheets rolled from non-ferrous metals [33]. As a result of the activation of shear bands, in the deformed material, a stronger grain refinement takes place in the areas of their presence, which improves the mechanical properties, compared with the sheets obtained in a symmetrical process [31].

The symmetrical rolling process is considered to be one in which all the process parameters are the same on both surfaces of the rolled band (the same parameters of friction, temperature, roll diameters, and their rotational speeds), presented in Figure 1a. In contrast, asymmetry can be introduced to the rolling process by using different roll materials (which generate a differentiation of friction conditions on the upper and lower surface of a deformed band or different temperatures on the upper and lower surface of the band), or by using different diameters or different rotational speeds of working rolls. Most often, rolling asymmetry process can be obtained by using rolls with different diameters at the same rotation speed of upper and lower rolls (Figure 1b) or with the differential rotation speed of upper and lower rolls and the same roll diameters (Figure 1c).

The asymmetry value can be determined by Equation (1) [34]:(1)$$a_{v}=\frac{D_{U}}{D_{L}}=\frac{\omega_{U}}{\omega_{L}}$$
where *D_U_*, *D_L_* indicate diameters of the upper and lower roll and *ω_U_*, *ω_L_* indicate the rotational speeds of the upper and lower rolls.

Literature reports indicate that plastic deformation under cryogenic conditions may significantly increase the strength parameters of precipitation-hardened aluminum alloys. In this study, the aim was to identify the effect of rolling technology (symmetry and asymmetry of the roll rotational speed) at room temperature and under cryogenic conditions on selected properties and the microstructure of the new version of the AA2519 alloy modified with scandium and zirconium.

## 2. Materials and Methods

The investigated material was a 4.75 mm thick AA2519-T62 alloy extrusion with the chemical composition given below (Table 1).

To investigate the rolling process of AA2519 alloy, a D150 two-high rolling mill (Czestochowa University of Technology, Czestochowa, Poland) was used with upper and lower roll diameters of 150 mm and a roll barrel length of 170 mm. Each roll had a separate drive from an asynchronous alternating current (AC) motor with a nominal power of 7.5 kW, through the reduction gear with a velocity ratio of 1:22.4 and a transmission shaft, which allowed a roll circumferential speed asymmetry to be introduced. The engine was controlled by the frequency converter ACS-601 manufactured by ABB Industry (Västerås, Sweden). A diagram of the laboratory stand is illustrated in Figure 2.

In the literature on the processes of asymmetric rolling of aluminum alloys, authors used various asymmetry coefficients up to 2.5 [35,36,37,38], but there are also examples of very large asymmetry coefficients of 4.0 [39]. In the work on the basis of the literature data as well as the earlier studies of the authors concerning the asymmetric rolling processes [33,34], the asymmetry coefficients ranging from 1.2 to 1.6 were used, which should reveal the influence of the asymmetry on the properties of the rolled material. The rolling processes in symmetry and asymmetry of the AA2519 alloy were conducted in four variants: variant I—symmetry rolling process *a_v_* = 1.0 (N1 and C1); variant II—asymmetry rolling process *a_v_* = 1.2 (N1.2 and C1.2); variant III—asymmetry rolling process *a_v_* = 1.4 (N1.4 and C1.4) and variant IV—asymmetry rolling process *a_v_* = 1.6 (N1.6 and C1.6), where N is rolling at room temperature and C is cryorolling. The samples produced in the rolling process are set out in Table 2. The rolling speed of 0.2 m/s was adopted (for the symmetry rolling). The AA2519 samples with dimensions of 4.75 mm × 50 mm × 150 mm were used for the rolling process. Rolling reduction equaled 20% in the first pass and 18% in the second pass, and the final thickness of the samples was equal to 3.14 mm.

The microstructure analysis involved a digital light microscope (LM) Olympus LEXT OLS 4100. The used etchant was Keller’s reagent, consisting of 20 mL H_2_O, 5 mL 63% HNO_3_, 1 mL 40% HF, and one drop of 36% HCl. The analysis was also supported by the Vickers microhardness distribution performed on the Struers DURA SCAN 70 microhardness tester, applying a 0.98 N load. For each sample, three lines of measurements were taken, from the top surface to a distance of 3.5 mm.

The microstrains were revealed, taking into account the X-ray patterns collected for the studied samples. The X-ray measurements were carried out using a Bruker D8 Advance diffractometer with CuKα radiation and an ultrafast semiconductor LynxEye detector. In order to reveal microstrains, the Willimson-Hall analysis was used [40].

The AA2519 in the initial state and after cryorolling (C1.6) was subjected to microstructure analysis using scanning transmission electron microscopy (STEM, Hitachi S-5500N) at 30 kV. Microscopic examinations were carried out on sample surfaces oriented parallel to the rolling direction. Specimens were prepared by ion-beam thinning using the Hitachi NB-5000 dual beam system (FIB/SEM). Firstly, in a standard sample-cutting procedure, a thin protective platinum layer was deposited on the sample surface and a three-sided trench around the platinum layer was prepared. Then, a lamella was removed from the sample by a nanomanipulator attached to the platinum layer. Next, the lamella was fixed to a copper holder by platinum deposition. The last step of the TEM lamella preparation was thinning to the final thickness of about 100 nm. In selected samples, a precise microstructural analysis was carried out.

## 3. Results and Discussion

### 3.1. Microhardness Distribution

For the main purpose of this investigation—to examine possibilities to improve the strength of AA2519-T62—the initial study concerned microhardness distribution, which allows the increase in strength to be evaluated and the homogeneity of this increase to be checked as well. The established microhardness distributions are presented below (Figure 3 and Figure 4).

AA2519-T62 alloy in the initial state has a microhardness value of 148.7 ± 5.3 HV0.1. By analyzing the obtained microhardness distributions, it can be concluded that both rolling at room temperature (Figure 3) and under cryogenic conditions (Figure 4) had a significant impact on the hardening of the AA2519-T62 alloy. However, the uniformity and values of the hardening significantly differ depending on the adopted rolling variant. The reason why some of the values do not precisely overlap are slight differences in the thickness of the samples after the rolling process. In the case of samples rolled at room temperature (Figure 3), a relatively uniform increase in microhardness was obtained over the entire section of the processed elements, with a slight increase in the middle part of the samples. The highest degree of strain hardening was reported for sample N1.0, with microhardness values reaching 183–187 HV.01, which corresponds to an increase in hardening by approx. 25%. The lowest hardening values were found in sample N1.4, rarely exceeded the value of 170 HV0.1 and were located at the value of 164 HV0.1, corresponding to a 10% increase in microhardness. In the case of samples N1.2 and N1.6, a relatively similar increase in the strengthening of approx. 15% was recorded. In the case of the samples rolled at room temperature, the impact of the asymmetry ratio on the microhardness is more visible than under cryogenic conditions. However, generally, the increase in the asymmetry of the process leads to higher shear deformation of the deformed samples, the highest microhardness values were obtained for the sample N1.0, rolled in the symmetry process (Figure 3). This variance can be caused by relatively close values of asymmetry, which differ only slightly in terms of affecting the grain sizes of aluminum alloys [41].

Analyzing the microhardness distributions in samples rolled under cryogenic conditions, significant discrepancies with regard to rolling at room temperature can be noticed (Figure 4). The changes in the C1.2. sample microhardness are noteworthy, for the value is the highest at the top surface (185 HV0.1) and then decreases, reaching a value almost close to the value of the base material (153 HV0.1). This indicates a significant heterogeneity of the strengthening, which is a significant problem from the point of view of using the product of this process as a construction material. The quite unexpected results are the hardening values recorded for the remaining samples. At a depth of up to 2 mm, the microhardness in sample C1.0 is generally around 172 HV0.1, which corresponds to an increase of only 15%. In the case of samples C1.4 and C1.6, the obtained hardening is even lower, amounting to approx. 10%.

The reason why rolling at room temperature gives higher hardening than cryorolling cannot be directly pointed out. Nevertheless, a possible explanation is the non-uniform deformation of the rolled samples. Plastic deformation in a whole volume of the material at room temperature is easier to achieve, resulting in strain hardening affecting the entire cross-section [42,43], as reflected in the microhardness distribution in the C1.2 sample (Figure 4). The situation where each layer of AA2519-T62 during cryorolling achieves different strain hardening may be considered a potential technological problem and requires further development of this technology.

### 3.2. Light Microscopy Observations

For microstructure analysis, two types of samples were selected: N1.6 and C1.6, characterized by relatively low but homogeneous hardening (Figure 5).

By analyzing the obtained microstructure images, it can be concluded that both samples are characterized by a banded, textured grainy structure (Figure 5a,b). The grains visible for the sample rolled at room temperature are randomly distributed in the matrix. However, in the case of the cold-rolled sample, the grains are formed in some needle agglomerates. In general, sample C1.6 has a more deformed structure than sample N1.6, which is reflected in smaller grain size. The structure is visible at higher magnifications, which allow the grain size of sample N1.6 to be estimated at 50.6 ± 27.7 μm and sample C1.6 at 31.1 ± 16.8 μm (Figure 5c,d). Considering that the grain size of the base material equals 90.8 ± 58.9 μm, it can be established that rolling with the asymmetry ratio of 1.6 caused a decrease in grain size by 44.3% and 65.7% under room-temperature and cryogenic conditions, respectively. It should be noted that the images were taken in the central part of each of the samples (about half of the thickness). These results, to some extent, contradict the theory of a lower plastic deformation of the inner layers of samples rolled under cryogenic conditions. The primary factors influencing the strengthening of the AA2519 alloy are grain size and the participation of the strengthening phase. The impact on the strengthening is greater in the case of the strengthened phase than in the case of the grain boundaries [11]. This may indicate that the evolution of the strengthening phase may result in a higher increase in the strengthening in rolling at room temperature. Since AA2519 is often used in an underaged condition (prior to the peak strength) to provide higher ductility, the potential explanation of the obtained results is the additional aging caused by heat generated during the rolling process [44]. Besides the temperature, another factor is plastic deformation, which induces the aging process [45]. As with all diffusion-based processes, the kinetics of aging strongly depend on temperature, and for this reason, it is harder to achieve in the case of cryorolled samples, where very low temperatures inhibit potential changes [46].

### 3.3. Microstrains

The microstrains detected in the studied sample could be caused, i.e., by external forces, and are observed as a significant broadening of reflexes in the X-ray diffraction pattern (*β_hkl_*). The average crystalline size can be calculated using Scherrer’s equation:(2)$$D=\frac{K\lambda }{\beta_{hkl}\cos\theta }$$
where: *D* indicates average crystalline size, *K* indicates shape factor, *λ* indicates wavelength of CuKα radiation, and *θ* indicates diffraction angle.

The strain (*ε*) induced in the material by external forces was calculated using the following relation:(3)$$\varepsilon =\frac{\beta_{hkl}}{4\tan\theta }$$

Taking into account that the particle size and strain contributions to line broadening are independent of themselves, the broadening of the line could be written as the sum of Relations (2) and (3):(4)$$\beta_{hkl}=\frac{K\lambda }{D\cos\theta }+4\varepsilon \tan\theta$$

Relation (4) could be rewritten in the following form, known as the Williamson–Hall equation:(5)$$\beta_{hkl}\cos\theta =\frac{K\lambda }{D}+4\varepsilon \sin\theta$$

During our analysis, Relation (5) was used to draw the Williamson–Hall plot. The strains were revealed by linear fitting. All fittings were higher than 0.93. Figure 6 presents microstrains revealed for all studied samples.

The microstrains induced in the base sample are clearly comparable with those observed in pure Al. The highest values of microstrains were detected for sample N1.6 and were caused by unequal rolling forces (asymmetric rolling). Moreover, a significant decrease in microstrain values was detected for cryorolled samples. As is well known, the temperature is the mean kinetic energy of some particles. In the solid state, the atoms oscillate around their equilibrium position in the lattice. These oscillations are strongly dependent on temperature. A decrease in temperature causes a decrease in the lattice constant in Al and its alloys. According to this phenomenon, the atoms at liquid nitro temperature (77K) are packed more closely and the rolling does not cause induction of such high strains as in samples at room temperature, i.e., N1.0 or N1.6. It can be stated that an increase in the rolling asymmetry ratio contributes to the higher values of microstrains and that this correlation takes place regardless of rolling conditions. At the same time, the cryorolled alloy is characterized by significantly lower values of microstrains. The microstrains of the sample are almost identical to the base material’s value (Figure 6). An additional factor that can influence the registered values of microstrains are aging processes that take place during deformation and that affect the crystal lattice parameter of Al-Cu alloys [47]. The obtained results can partially prove the potential additional aging caused by heat generated during the rolling process at room temperature [44,45].

### 3.4. TEM Observations

The microstructure of the base material and sample C1.6, observed in TEM, are presented on the next page (Figure 7).

The microstructure of the base material contains numerous precipitates of different shapes (Figure 7a). The most common type of precipitate is spherical Al-Cu-Mg precipitates below 0.5 μm in size [48,49,50,51]. The single rod-like structure observed in Figure 7a is most probably an Al-Cu-Mn precipitate, identified in similar investigations [48,49]. The boundaries of grains and precipitates are very clear and free of dislocations (Figure 7a). Comparing obtained images, it can be stated that the sample subjected to the rolling under cryogenic conditions is characterized by higher dislocations density (Figure 7b) than the base material (Figure 7a). The cryorolling process resulted in the formation of a dislocation cell structure with dense dislocation walls of about 0.1–0.2 μm in size (Figure 7b). The size of the cells is uneven. It is clear that high densities of dislocations were trapped between the spherical precipitates as the result of cold plastic deformation (Figure 7b). The high accumulation of dislocations close to precipitates gives a relatively small size of dislocation cells [50]. Similar dislocation structures have been reported in the literature in some nonequilibrium deformation processes, including cryorolling of 2XXX alloys [22,52]. Inside the dislocations cells, the low-density (light) dislocation clusters can be observed (Figure 7b) [53,54].

## 4. Conclusions

This research aimed to study the influence of asymmetry rolling under different conditions on AA2519-T62. The obtained results contribute to the state of the art on cryoforming of metallic materials, providing information on microhardness distribution, microstrains, and microstructural effects of deformation. These data can be applied to industrial practices to develop the process of rolling AA2519-T62 (Sc, Zr) to simultaneously form and increase the hardness of the alloy. The most significant conclusions that can be drawn from this investigation are:

Both rolling at room temperature and under cryogenic conditions increase the mi-crohardness of AA2519-T62 by at least 10%. The highest reported increase (25%) was obtained for samples rolled at room temperature in a traditional symmetry rolling process.Rolling with the asymmetry ratio of 1.6 caused the decrease in grain size by 44.3% and 65.7% under room-temperature and cryogenic conditions, respectively. Despite the more severe grain refinement, samples rolled under cryogenic conditions are characterized by a lower increase in microhardness than samples rolled at room temperature.The application of rolling with the asymmetry ratio contained within the range of 1.2–16 only slightly affected the microhardness values of the samples rolled at room temperature and under cryogenic conditions.The strains induced in the base sample were comparable with the one detected for pure Al. The asymmetric rolling caused an induction of the highest strains in the studied samples. Moreover, the cryorolling process is characterized by the induction of significantly lower microstrains than those observed for samples rolled at room temperature.The cryorolling process resulted in the formation of the dislocation cell structure with dense dislocation walls of about 0.1–0.2 μm in size. A high accumulation of dislocations is reported in the surrounding of the spherical precipitates.

## Figures and Tables

**Figure 1 materials-15-07712-f001:**
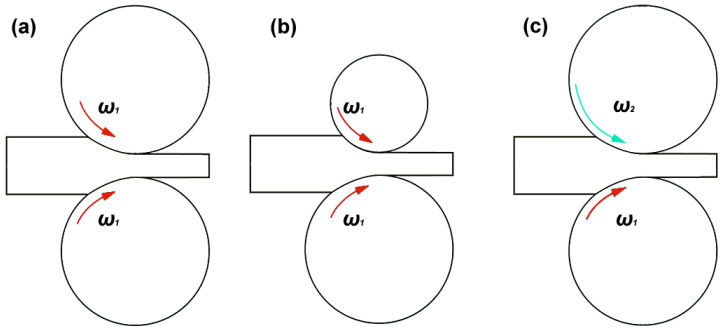
A schematic drawing of: (**a**) symmetry rolling process, (**b**) asymmetry with different roll diameters, (**c**) asymmetry with a differential rotation speed of rolls.

**Figure 2 materials-15-07712-f002:**
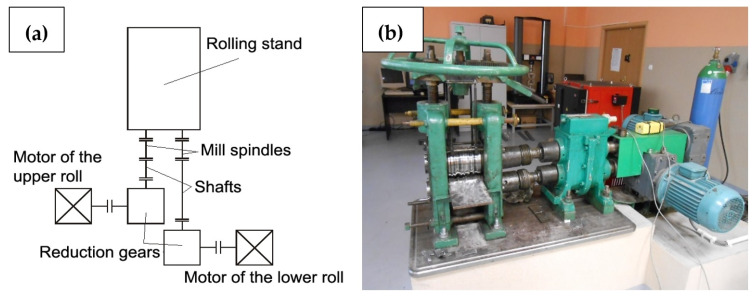
The layout of the D150 two-high rolling mill (**a**) and a general view (**b**) [34].

**Figure 3 materials-15-07712-f003:**
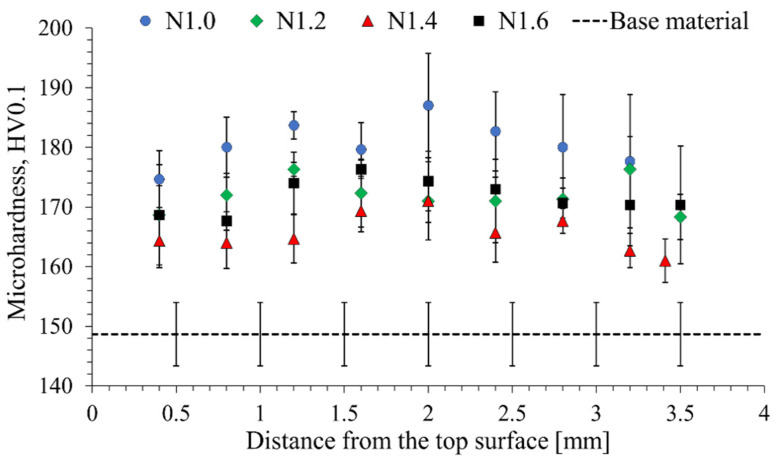
Distribution of microhardness in the samples rolled at room temperature.

**Figure 4 materials-15-07712-f004:**
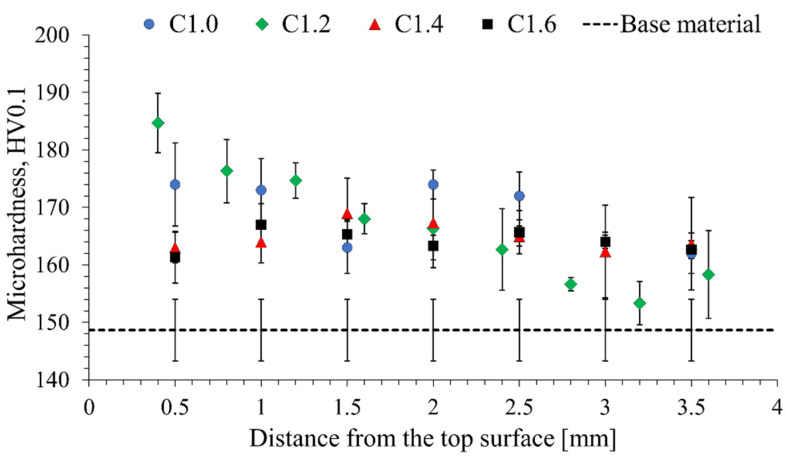
Distribution of microhardness in the samples rolled under cryogenic conditions.

**Figure 5 materials-15-07712-f005:**
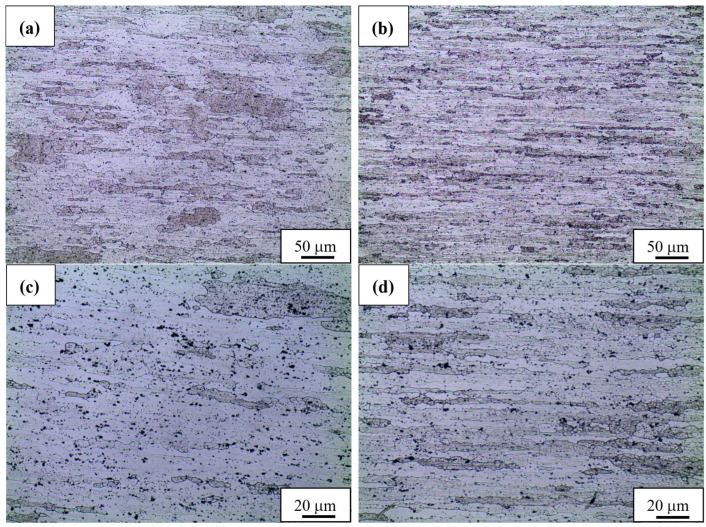
Microstructure of the N1.6 (**a**,**c**) and C1.6 sample (**b**,**d**) with different magnifications.

**Figure 6 materials-15-07712-f006:**
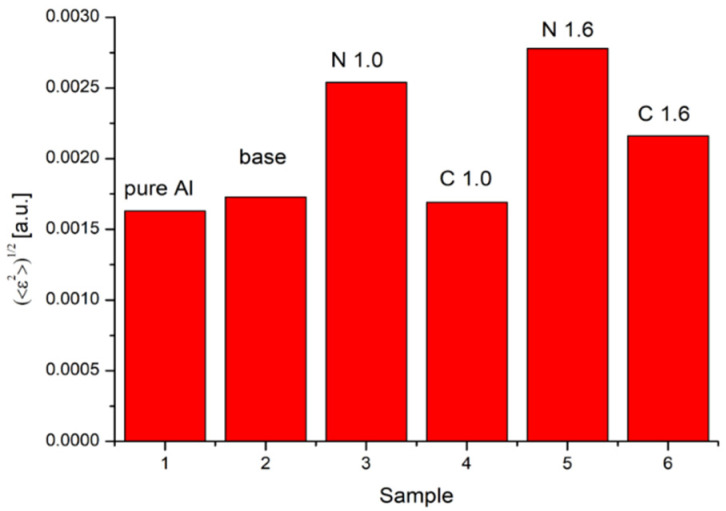
Microstrains calculated for all studied samples.

**Figure 7 materials-15-07712-f007:**
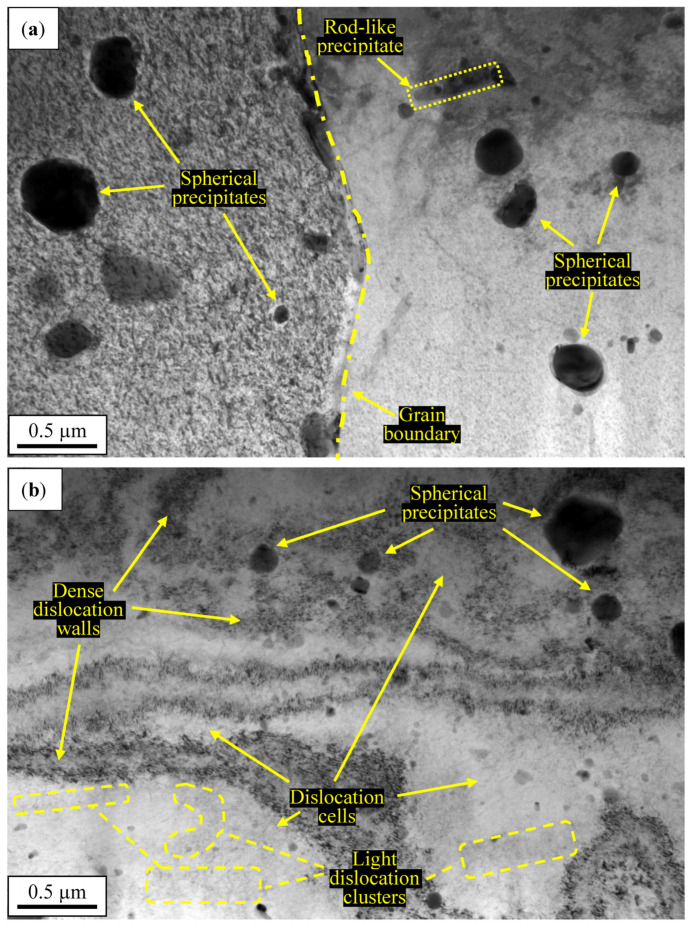
TEM images of the base material (**a**) and sample C1.6 (**b**).

**Table 1 materials-15-07712-t001:** Chemical composition of AA2519 alloy.

Fe	Si	Cu	Zn	Ti	Mn	Mg	Ni	Zr	Sc	V	Al
0.11	0.08	6.32	0.05	0.08	0.17	0.33	0.02	0.19	0.16	0.10	Base

**Table 2 materials-15-07712-t002:** Pass schedule of the rolling process.

Variant of Rolling/Asymmetry Ratio *a_v_*	20 °C	−196 °C
I/1.0 (symmetry rolling)	N1.0	C1.0
II/1.2	N1.2	C1.2
III/1.4	N1.4	C1.4
IV/1.6	N1.6	C1.6

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
