# Peer review of "Study on Symmetry and Asymmetry Rolling of AA2519-T62 Alloy at Room-Temperature and Cryogenic Conditions"

_materials, 2022, doi:10.3390/ma15217712_

Round 1

Reviewer 1 Report

Dear Authors,

Your contribution is of high quality and is characterized by high applicability in industrial practice. The forming of aluminum alloys usable for applications in military technology is currently very topical. I have a few comments/recommendations about your article:

1) In the sentence on lines 44-46 you mention the usefulness of Al alloy for the aerospace and military industries - can you give a specific example for military use?

2) Figure 6 - the given scale is hard to read, change it. Also mark the dislocation bands with an arrow, as it may not be obvious to the reader exactly where the dislocations are located.

3) The article lacks a discussion of the results, make a critical evaluation of your results with respect to the results obtained by other authors. State the significance of your study and the applicability of your results in industrial practice. In what areas can your results be applied? What are the advantages of the molding process you presented? Please complete before Conclusions.

4) The conclusion lacks a brief summary of the results achieved and what the authors contributed to in the given area.

Author Response

            We would like to thank the Reviewers for their time and effort in carefully checking our manuscript. Being very grateful for all valuable comments, insights, and advice, we have made the suggested changes in the manuscript and prepared answers for all comments.

Reviewer 2 Report

1. According to Figure 2, the error bar is too big to doubt the authenticity of its hardness. Is it this change trend?

2. The TEM description is too simple. What is the black spherical one? What is the composition of the precipitation phase? What conclusions or mechanisms do you draw from these microscopic shapes? This needs to be further improved!

Author Response

(The authors gave the same response as above.)

Reviewer 3 Report

The following points are necessary to address clearly for further processing;

[1]. The authors have highlighted the effect of room temperature and cryogenic rolling effect but forget to include the effect of symmetry and asymmetry effect of rolling. This is the major inclusion in the study as highlighted in the title and in the introduction section that spotted the novelty of this work.

[2]. To increase the readability, the authors need to define briefly the difference between symmetry and asymmetry rolling technology. Or include a pictorial diagram in the Materials and Method Section.

[3]. On what basis, this variant of rolling / asymmetry ratio av is selected? Include some relevant or near similar values/references in the introduction from other similar research articles.

[4].  Check Fig. 2, why the N1.4 value is not plotted at 3.5mm. Also check the values of N1.0 AT 0.5 and 3.5mm, whether it is missing or overlapping with other values.

[5].  The same is the case with Fig. 3, the hardness values of C1.2 is plotted outside the designated distance. Is it intentional?

[6].  Correct the caption of Fig. 4. Microstructure of the N1.6 (a,c) and C1.6 sample (a,d) with different magnifications. Both are indicated with “a”.

[7].  If C.16 is more deformed than N1.4, but higher values of hardness are depicted for room temperature deformation in Fig. 4 than Fig. 5, elaborate or support your statement on lines 210-211.

[8].  Any standard ASTM or any other standard procedure for Microstrains?

[9]. All fittings were higher than 0.93 (on line 246). Explain this. What happens if lower than this value or any other threshold value?

[10]. What is the importance of obtaining the Microstrains in this study? Relate the obtained results with the Hardness, Microstructure, and TEM results.

[11]. Highlight the dislocations in Fig. 6, Furthermore, the explanation is not enough here. Add explicit characterization for TEM results.

Author Response

(The authors gave the same response as above.)

Round 2

Reviewer 2 Report

no

Author Response

(The authors gave the same response as above.)

Reviewer 3 Report

Authors need to address the following points in relation to the comments of round 1;

Comment No. 1 is not addressed correctly at all. Authors need to highlight the effect of symmetry and asymmetry rolling in the abstract although they have included the effects of room temperature and cryogenic rolling only.

For Comments 4 and 5, include this reason somewhere under heading 3.1, "slight differences in the thickness of the samples is after the rolling process".

Author Response

(The authors gave the same response as above.)
